# Density-Controlled Growth of ZnO Nanowalls for High-Performance Photocatalysts

**DOI:** 10.3390/ma15249008

**Published:** 2022-12-16

**Authors:** Yu-Cheng Chang, Ying-Ru Lin, Sheng-Wen Chen, Chia-Man Chou

**Affiliations:** 1Department of Materials Science and Engineering, Feng Chia University, Taichung 407102, Taiwan; 2Department of Materials Science and Engineering, National Yang Ming Chiao Tung University, Hsinchu 30010, Taiwan; 3Department of Surgery, Taichung Veterans General Hospital, Taichung 40705, Taiwan; 4College of Medicine, National Yang Ming Chiao Tung University, Taipei 11221, Taiwan; 5Department of Post-Baccalaureate Medicine, National Chung Hsing University, Taichung 40227, Taiwan

**Keywords:** ZnO nanowires, ZnO nanowalls, aqueous solution method, UVC light, methylene blue, tetracycline, reusability

## Abstract

ZnO nanowires and nanowalls can be fabricated on the glass substrate with a ZnO seed film and low-cost aluminum (Al) foil by the aqueous solution method (ASM), respectively. The different concentrations of ZnO precursors can use to control the densities of ZnO nanowalls. In addition, FESEM, FETEM, EDS, XRD, XPS, and CL were used to evaluate the characteristics of ZnO nanowalls. The ZnO nanowalls exhibited higher photocatalytic efficiency (99.4%) than that of ZnO nanowires (53.3%) for methylene blue (MB) degradation under UVC light irradiation at the ZnO precursors of 50 mM. This result is attributed to ZnO nanowalls with Al-doped, which can improve the separation of photogenerated electron-hole pairs for enhanced photocatalytic activity. In addition, ZnO nanowalls can also reveal higher photocatalytic activity for the degradation of tetracycline capsules (TC) rather than commercial ZnO nanopowder under UVC light irradiation. The superoxide and hydroxyl radicals play essential roles in the degradation of MB and TC solutions by the radical-trapping experiment. Furthermore, the ZnO nanowalls exhibit excellent recycling and reuse capacity for up to four cycles for the degradation of MB and TC. This study highlights the potential use of ZnO nanowalls directly grown on commercial and low-cost Al foil as noble metal-free photocatalysis.

## 1. Introduction

Nanowalls are novel three-dimensional nanostructures that have attracted increasing attention due to their large surface-to-volume ratios and extremely thin wall thicknesses [1,2]. Various semiconductor nanowalls have been synthesized by chemical or physical routes, such as ZnO, ZnS, Cd(OH)_2_, and GaN [3,4,5,6]. ZnO is a unique n-type semiconductor material with a large exciton binding energy of 60 meV and a wide direct band gap energy of 3.37 eV, which has been extensively investigated for its promising applications [7,8,9,10]. Furthermore, ZnO is one of the ideal photocatalysts for degrading environmental pollutants, attributed to its non-toxicity, low cost, and high oxidative power of photogenerated holes [11,12,13,14,15]. Recently, ZnO nanowalls have been synthesized by vapor transport and conduction [16], pulsed laser deposition [17], electrochemical deposition [18], sonochemical method [19], and aqueous solution method (ASM) [1,2,20,21]. The ASM approach is the most successful fabrication method for growing ZnO nanowalls at low reaction temperatures with flexibility, simplicity, and cost-effectiveness, while being less hazardous, and environmentally friendly [22].

ZnO nanowalls have been widely grown on various substrates, such as glass [23], aluminum foil [1,24], silicon [25], polyethersulfone (PES) [26], Al alloy [27], and polyethylene terephthalate (PET) [28]. Among them, they use high-purity aluminum (Al) foil or deposit a high-purity Al film on the substrate to grow ZnO nanowalls [1,24,25]. However, fewer reports use commercial and low-cost Al foil to grow ZnO nanowalls directly. In addition, there is no report on growing ZnO nanowalls for photocatalytic degradation of tetracycline. Tetracycline is one of the main antibiotics used for human, pet, and agricultural purposes [29]. Among the different types of antibiotics, tetracycline needs more attention mainly because it exhibits severe environmental problems, including ecological risks and damage to human health [30].

Herein, we investigated the effect of ZnO nanowalls with different densities grown at different concentrations of ZnO precursors on the commercial and low-cost Al foil by the ASM approach. The commercial and low-cost Al foil is primarily used in the household kitchens. ZnO nanowalls grown on the Al foil exhibited higher photocatalytic activity and a reaction constant of 6.65 times higher than ZnO nanowires grown on the glass substrate at the same concentrations of ZnO precursors under UVC light irradiation. This result is attributed to the enhanced photocatalytic activity by inhibiting the recombination of photogenerated electron-hole pairs by the Al-doped ZnO nanowalls. In addition, ZnO nanowalls can also use for the photodegradation of TC under UVC light irradiation with high reusability. ZnO nanowalls can provide a facile, low-cost, high photocatalytic activity and reusability, which can benefit many other related fields.

## 2. Materials and Methods

### 2.1. Materials

An aluminum (Al) foil (7.62 m × 30.4 cm (2.1 USD) was commercially obtained from Diamond (North Billerica, MA, USA). All chemicals were purchased from commercial sources and used without further purification. Zinc nitrate dihydrate (98%, Alfa Aesar, Ward Hill, MA, USA), hexamethylenetetramine (HMTA, 99%, Alfa Aesar, USA), zinc acetate dihydrate (97%, Alfa Aesar, USA), ZnO nanopowder (Uniregion Bio-Tech, Taoyuan, Taiwan), methylene blue (MB, 95%, Alfa Aesar, USA), tetracycline capsules (TC, 250 mg, Veterans Pharmaceutical, Taoyuan, Taiwan), triethanolamine (TEOA, 98%, Alfa Aesar, USA), isopropyl alcohol (IPA, 99.5%, Alfa Aesar, USA), L-ascorbic acid (AA, 98%, Alfa Aesar, USA), silver nitrate (AgNO_3_, 99%, Alfa Aesar, USA) and ethanol (C_2_H_5_OH, 99%, Sigma-Aldrich, Darmstadt, Germany) were used in this experiments. De-ionized water with a resistivity higher than 18.2 MΩ was used for all solution preparations.

### 2.2. Synthesis of ZnO Nanowires

Glass substrates (1.5 cm × 2.5 cm) were thoroughly cleaned in 95% ethanol by an ultrasonic vibrator for 10 min to remove particles and organic contaminants from the substrate surface. Next, ZnO seed film was prepared by spin-coating a layer of ethanol solution with 20 mM zinc acetate dihydrate, followed by thermal annealing at 80 °C and 350 °C for 3 min and 20 min. Finally, the ZnO nanowires were directly synthesized on the glass substrates with a ZnO seed film by an ASM in a 100 mL aqueous solution with different ZnO precursors (equimolar zinc nitrate dihydrate and HMTA). The glass substrates with ZnO seed film (1.5 cm × 2.5 cm) were pasted on a glass sheet, placed in a sealed crystallizing dish (150 mL) containing the above reaction solution, and heated in a hotplate at T = 90 °C for 3 h. The reaction device of the growth of ZnO nanowires is illustrated in Figure 1.

### 2.3. Synthesis of ZnO Nanowalls

An Al foil was cut into a 1.5 cm × 2.5 cm substrate. These Al foils (1.5 cm × 2.5 cm) were thoroughly cleaned in 95% ethanol by an ultrasonic vibrator for 10 min to remove particles and organic contaminants from the substrate surface. Then, the ZnO nanowalls were directly synthesized by an ASM in a 100 mL aqueous solution with different ZnO precursors (equimolar zinc nitrate dihydrate and HMTA). The Al foils were pasted on a glass sheet, placed in a sealed crystallizing dish (150 mL) containing the above reaction solution, and heated in a hotplate at T = 90 °C for 3 h. The reaction device of the growth of ZnO nanowalls is illustrated in Figure 1.

### 2.4. Material Characterization

The surface morphologies and the crystal structures of the ZnO nanowalls were investigated using field-emission scanning electron microscopy (FESEM, Hitachi S4800, Kyoto, Japan), field-emission transmission electron microscopy (FETEM, JEOL-2100F, Kyoto, Japan), and X-ray diffraction (XRD, Bruker D2 phaser, USA). X-ray photoelectron spectroscopy (XPS, ULVAC-PHI PHI 5000 Versaprobe II system, Kanagawa, Japan) was used to identify the surface elemental composition and electron configuration of ZnO nanowalls grown on the Al foil. Finally, the cathodoluminescence (CL) of the ZnO nanowalls was evaluated using a CL spectrometer (JEOL, JSM7001F, Japan).

### 2.5. Photocatalytic Activity Test

The photocatalytic activities of as-synthesized photocatalysts were evaluated via the degradation of MB (10^−5^ M) and TC (10^−4^ M) aqueous solutions. A UV lamp (253.7 nm, 10 W, Philip, Amsterdam, The Netherlands) was used as a UVC light source in a typical photocatalytic process. The concentrations of MB and TC solutions were measured using a DR/UV-Vis spectrometer (Hitachi U-2900, Tokyo, Japan) to record changes in characteristic absorption bands. The photocatalytic efficiency of photocatalysts under UVC light irradiation was determined by C/C_0_, where C_0_ and C were the initial and instantaneous concentrations of MB or TC solutions, respectively.

## 3. Results

Figure 2a displays the fabrication processes of ZnO nanowires by an ASM. First, the glass substrate was grown with a ZnO seed film by combining spin coating and thermal annealing processes. Second, ZnO nanowires were synthesized on the glass substrate with a ZnO seed film by an ASM at the reaction temperature of 90 °C for 3 h under different ZnO precursor concentrations. Figure 2b–e display the top-view FESEM images of well-aligned ZnO nanowires directly synthesized on the glass substrate with a ZnO seed film from different concentrations of ZnO precursors. The concentrations of zinc precursors are 10, 20, 50, and 75 mM, respectively. The average diameters of ZnO nanowires are 49.5, 65.2, 82.1, and 165.3 nm, respectively. The average lengths of ZnO nanowires are 1.75, 1.89, 2.01, and 2.29 μm, respectively. The average diameters and lengths of ZnO nanowires tend to increase with the increase in ZnO precursor concentration, as shown in Appendix A. This result shows that the ZnO precursor concentration can be used to control the sizes of ZnO nanowires.

Figure 3 shows the FETEM characterization results of ZnO nanowires grown at the concentrations of ZnO precursors of 50 mM. Figure 3a displays the FETEM image of a ZnO nanowires with a diameter of 86.2 nm. The selected area electron diffraction (SAED) pattern (Figure 3b) exhibits the (010) zone axis of hexagonal ZnO (JCPDS No. 75–0576). Figure 3c shows the HRTEM image of ZnO nanowires with a lattice spacing of 0.260 nm, corresponding to the (002) crystal plane of hexagonal ZnO (JCPDS number 75–0576). Based on the above SAED pattern and HRTEM image, the ZnO nanowire is single crystalline for growing along the [001] direction. Figure 3d displays the energy dispersive spectroscopy (EDS) spectrum of the ZnO nanowire in Figure 3a. It can be observed that Zn and O determine the composition of ZnO nanowires. The composition of Cu is ascribed to the TEM grid. The XRD diffraction pattern of ZnO nanowires grew on the glass substrate with a ZnO seed film at the concentrations of ZnO precursors of 50 mM. An intense and sharp diffraction peak corresponding to the (002) crystal plane of hexagonal ZnO (JCPDS No. 75–0576) suggests that the preferred growth direction of ZnO nanowires is the [001] direction.

Figure 4 displays 45° tilt-view FESEM images of ZnO nanowalls that were directly grown on the Al foils from the different concentrations of ZnO precursors by an ASM at the growth temperature of 90 °C for 3 h. In general, HMTA can fix the pH value of the solution at around 6 and react with the Al substrate to form hydroxyl ions of Al(OH)_4_^−^. The Al(OH)_4_^−^ binding to the Zn^2+^ terminal surface prevents ZnO growth along the (001) direction and promotes lateral growth to form nanowalls [1,31]. The concentrations of ZnO precursors are 10, 20, 50, and 75 mM, respectively. The density of the ZnO nanowalls tends to increase with the concentrations of the ZnO precursors. This phenomenon shall be ascribed to the higher concentrations of ZnO precursors, which accelerate the reaction and are more favorable for the growth of ZnO nanowalls on the Al foil [32].

Figure 5 displays the XRD diffraction patterns of (a) Al foil and (b) ZnO nanowalls grown at the ZnO precursors of 50 mM. For Al foil, the diffraction peaks at 40.3° and 44.7° correspond to (004) and (200) crystal planes, confirming the orthorhombic Al_2_O_3_ (JCPDS No. 88–0107) and cubic Al (JCPDS No. 89–4037), respectively. For ZnO nanowalls, there are another five diffraction peaks at 31.7°, 34.4°, 36.3°, 47.5°, and 56.6° corresponding to (100), (002), (101), (102), and (110) crystal planes, proving the growth of hexagonal ZnO crystal phase (JCPDS No. 75–0576). This XRD result confirms that the ZnO nanowalls do not contain any other impurities.

Figure 6a displays the FETEM image of ZnO nanowalls grown at the concentrations of ZnO precursors of 50 mM. The ZnO nanowall is composed of many small grains. The SAED pattern (Figure 6b) indicates that the ZnO nanowall is polycrystalline and consistent with hexagonal phase ZnO (JCPDS no. 75–0576). Figure 6c shows the HRTEM image of a ZnO nanowall with two lattice spacing of 0.248 and 0191 nm corresponding to the (101) and (102) crystal planes of the hexagonal phase ZnO (JCPDS no. 75–0576). Figure 6d,e show corresponding EDS elemental mapping images and spectrum of ZnO nanowall for Zn, O, and Al, respectively. It can be seen that Zn, O, and Al determine the composition of ZnO nanowall. The content of Al is determined to be 11.3 at%. This result can also verify that Al has been doped into the ZnO nanowalls.

XPS can be used to analyze the surface composition and chemical state of ZnO nanowalls grown at the ZnO precursors of 50 mM. The survey scan spectrum confirms the presence of Zn, O, and Al elements, as shown in Figure 7a. The presence of C 1s may come from the organic layer decorated on the ZnO nanowall surface or the pump oil in the vacuum system of the XPS apparatus. The high-resolution Zn 2p spectrum (Figure 7b) shows that two peaks at 1021.7 and 1044.8 eV corresponding to the Zn 2p_3/2_ and Zn 2p_1/2_, respectively. The asymmetric O 1s peak (Figure 7c) can be divided into three sub-peaks at 529.7, 530.8, and 532.3 eV, corresponding to O species in the lattice (O_L_), oxygen vacancies or defects (O_V_) and chemisorption or dissociation (O_C_), respectively. It can be observed that the oxygen vacancies or defects peak is significantly stronger than the lattice peak. This result proves that the Al substitute for Zn sites in ZnO nanowalls shall induce the appearance of oxygen vacancies or defects [33]. In addition, the ZnO nanowalls also exhibited significantly higher chemisorption or dissociation of O species. The high-resolution Al 2p spectrum (Figure 7d) shows that one peak at 74.4 eV corresponds to the reported values for Al_2_O_3_: Al 2p at 74.35 eV [3,34]. The Al peak in ZnO nanowalls indicates that the Al element has been successfully doped into the ZnO lattice.

The photocatalytic degradation of MB further evaluated the photocatalytic activities of as-synthesized ZnO nanowires and ZnO nanowalls grown at different concentrations of ZnO precursors under UVC light irradiation. Monitor the temporal change in MB concentration by examining the change in the maximum absorbance in the UV–vis spectrum at 664 nm. Figure 8a shows the degradation rate of ZnO nanowires grown at different concentrations of ZnO precursors. The degradation rate of ZnO nanowires is 36.3 (10 mM), 40.6 (25 mM), 53.3 (50 mM), and 50.9% (75 mM), respectively. ZnO nanowires (50 mM) revealed the highest photocatalytic activity in the decomposition of MB. The kinetics of the photocatalytic degradation reaction can be fitted to a pseudo-first-order reaction, and the photodegradation rate constants (k, min^−1^) for ZnO nanowires grown at different concentrations of ZnO precursors are estimated from the slopes of the straight lines, as shown in Figure 8b. The rate constants of ZnO nanowires with different concentrations of ZnO precursors can be calculated as 0.00238 (10 mM), 0.00292 (25 mM), 0.00444 (50 mM), and 0.00414 (75 mM) min^−1^, respectively.

Figure 8c reveals the degradation rate of ZnO nanowalls grown at different concentrations of ZnO precursors. The degradation rate of ZnO nanowalls is 67.3 (10 mM), 85.7 (25 mM), 99.4 (50 mM), and 94.9% (75 mM), respectively. The photodegradation rate of MB followed the order of the ZnO precursors concentrations of ZnO nanowalls by 50 mM > 75 mM > 25 mM  > 10 mM. The photocatalytic efficiency of ZnO nanowalls is in comparison with other literature (Table 1), such as γ-Fe_2_O_3_/Fe_3_O_4_/SiO_2_ photocatalysts [35], Bi/BaSnO_3_@HNTs nanomaterials [36], Cu-doped ZnO nanoneedles [33], and Bi_2_WO_6_/ZIF8 photocatalysts [37]. Figure 8d shows the pseudo-first-order linear relationship of ZnO nanowalls grown at different concentrations of ZnO precursors. The rate constants of ZnO nanowalls with different concentrations of ZnO precursors can be calculated as 0.00636 (10 mM), 0.01149 (25 mM), 0.02957 (50 mM), and 0.01721 (75 mM) min^−1^, respectively. For the photocatalytic degradation of MB, the photocatalytic efficiency of ZnO nanowalls (50 mM) is about 6.65 times higher than ZnO nanowires (50 mM). This result suggests that ZnO nanowalls can greatly facilitate their practical applications to eliminate various environmental pollutants in wastewater.

Figure 8e shows the CL spectra of ZnO nanowires and ZnO nanowalls that grew at the concentrations of ZnO precursor of 50 mM. A strong UV emission (near-band-edge emission) and a weak visible emission (deep-level emission) are observed at 380 nm (3.26 eV) and 560 nm (2.21 eV), respectively, for ZnO nanowires. A weak UV emission (near-band-edge emission) and a strong visible emission (deep-level emission) are observed at 380 nm (3.26 eV) and 562 nm (2.21 eV), respectively, for ZnO nanowalls. The ZnO nanowires exhibited a stronger emission property (near-band-edge emission or deep-level emission) than the ZnO nanowalls. This phenomenon is attributed to ZnO nanowall with doped Al, which can effectively separate photogenerated electron-hole pairs for improved photocatalytic activity.

To demonstrate that ZnO nanowalls can also be used for photocatalytic degradation of drugs. Herein, we chose the tetracycline capsule (TC) as one of the drugs. Figure 9a reveals the degradation rate of ZnO nanowalls (50 mM) and commercial ZnO nanopowder under UVC light irradiation. The degradation rate of ZnO nanowalls and ZnO nanopowder is 82.6 and 73.9%, respectively. The pseudo-first-order linear relationship of ZnO nanowalls and ZnO nanopowder is shown in Figure 9b. The corresponding reaction constants of TC degradation over ZnO nanowalls and ZnO nanopowder can be calculated as 0.01007 and 0.00724 min^−1^, respectively. ZnO nanowalls exhibited higher photocatalytic activity and a reaction constant of 1.39 times higher than ZnO nanopowder under UVC light irradiation.

The reusability of ZnO nanowalls was studied by recycling experiments with MB and TC solution under UVC light irradiation, as shown in Figure 10. For the MB solution (Figure 10a), the photocatalytic efficiency was 99.8, 95.0, 96.5, and 97.9% for the four cycles. For the TC solution (Figure 10b), the photocatalytic efficiency was 82.3, 80.0, 76.6, and 74.8% for the four cycles. After four cycles, the photocatalytic efficiency of ZnO nanowalls exhibited insignificant changes. ZnO nanowalls revealed a long lifetime as photocatalysts with high activity and reusability. In addition, ZnO nanowalls were synthesized directly on Al foil, thus simplifying the recycling process and making ZnO nanowalls with stable and economical photocatalysts.

In general, the photocatalytic degradation process involves a variety of active species, including hydroxyl radicals (·OH), superoxide radicals (·O_2_^−^), photogenerated electrons (e^−^), and photogenerated holes (h^+^) [38]. Herein, four kinds of scavengers were used to explore the photocatalytic mechanism of ZnO nanowalls for MB and TC under UVC light irradiation, as shown in Figure 11a,b. In order to distinguish the role of active species in MB or TC degradation and explain the reaction mechanism, TEOA, IPA, AgNO_3_, and AA were selected as quenchers for h^+^, ·OH, e^−^, and ·O_2_^−^, respectively [39,40,41]. It can be observed that the introduction of IPA and AA scavengers in the photocatalytic reaction results in a significant decrease in the photocatalytic efficiency. These results prove that ·OH and ·O_2_^−^ play the main active radicals in the photodegradation of MB or TC. Based on the above studies, the possible photocatalytic mechanism of ZnO nanowalls is shown in Figure 11c. The oxygen vacancy can create a new electric state band at the bottom of the CB of ZnO nanowalls. Herein, the photogenerated electrons of ZnO nanowalls can be excited from their VB into CB or oxygen vacancy under UVC light irradiation. The photogenerated electrons (e^–^) can reduce O_2_ molecules to ·O_2_^−^ for MB or TC solution degradation. The photogenerated holes (h^+^) in the VB of ZnO nanowalls can react with H_2_O molecules to form hydroxyl radicals (·OH) for MB or TC solution degradation.

## 4. Conclusions

In this study, we report a facile ASM that can be fabricated directly on commercial and low-cost Al foil without ZnO seed film at a low reaction temperature of 90 °C. The concentration of ZnO precursor plays a substantial role in influencing the density of ZnO nanowalls. The morphology, microstructures, crystal structures, chemical structures, and optical properties of ZnO nanowalls were investigated by FESEM, X-ray diffraction, FETEM, EDS, XPS, and CL. Compared with ZnO nanowires, the photocatalytic efficiency of ZnO nanowalls is notably boosted from 53.3 to 99.4% in methylene blue degradation under UVC light irradiation. This phenomenon is ascribed to the effective inhibition of the recombination of electron-hole pairs via Al-doped. In addition, ZnO nanowalls can also exhibit higher photocatalytic activity than ZnO nanopowder for treating TC under UVC light irradiation. Furthermore, the reusability test can demonstrate that the ZnO nanowalls on the Al foil still maintain high photocatalytic activity for the degradation of MB and TC, even after four cycles, without any significant decline. ZnO nanowalls can provide a simple, low-cost, high photocatalytic efficiency and reusability, which can benefit many other related fields.

## Figures and Tables

**Figure 1 materials-15-09008-f001:**
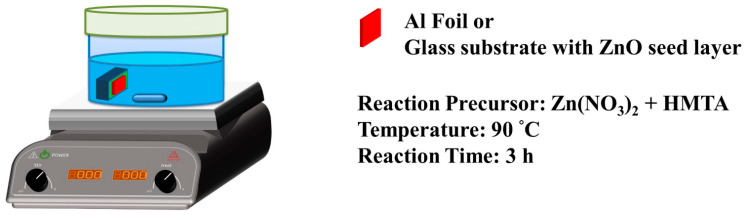
Schematic diagram of the reaction device for the growth of ZnO nanowalls.

**Figure 2 materials-15-09008-f002:**
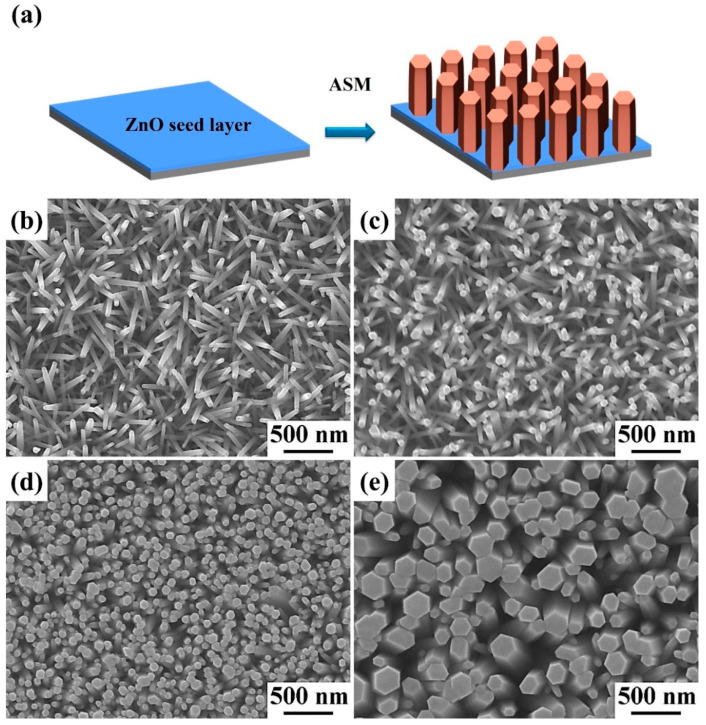
(**a**) Schematic diagram of the reaction processes for the growth of ZnO nanowires. (**b**–**e**) The top-view FESEM images of ZnO nanowires synthesized on the glass substrate with a ZnO seed film at different concentrations of ZnO precursors. The concentrations of ZnO precursors are (**b**) 10, (**c**) 25, (**d**) 50, and (**e**) 75 mM, respectively.

**Figure 3 materials-15-09008-f003:**
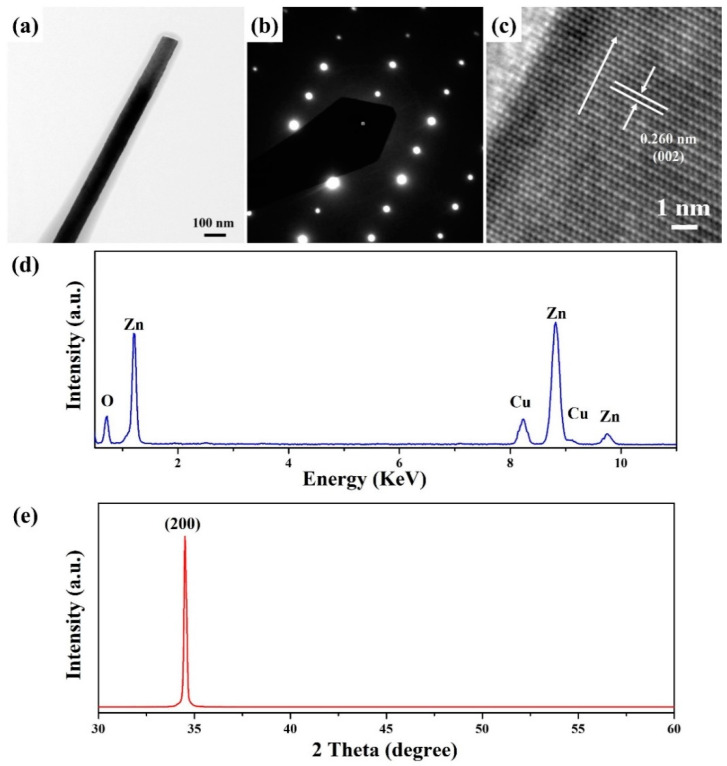
(**a**) FETEM, (**b**) SAED pattern, (**c**) HRTEM images, and (**d**) EDS spectrum of a ZnO nanowire (50 mM). (**e**) The XRD diffraction pattern of ZnO nanowires (50 mM) synthesized on the glass substrate with a ZnO seed film.

**Figure 4 materials-15-09008-f004:**
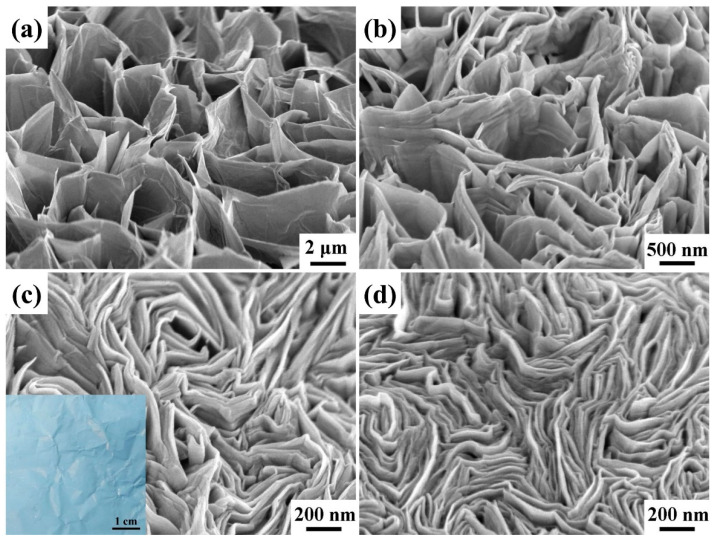
The tilt-view FESEM images of ZnO nanowalls grown on the Al foil at different concentrations of ZnO precursors. The concentrations of ZnO precursors are (**a**) 10, (**b**) 25, (**c**) 50, and (**d**) 75 mM, respectively.

**Figure 5 materials-15-09008-f005:**
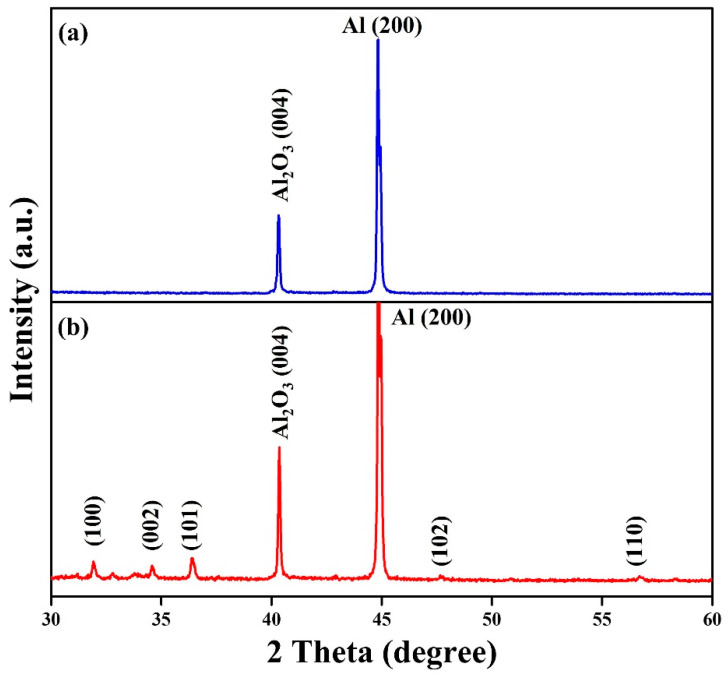
XRD diffraction pattern of (**a**) Al foil and (**b**) ZnO nanowalls (50 mM).

**Figure 6 materials-15-09008-f006:**
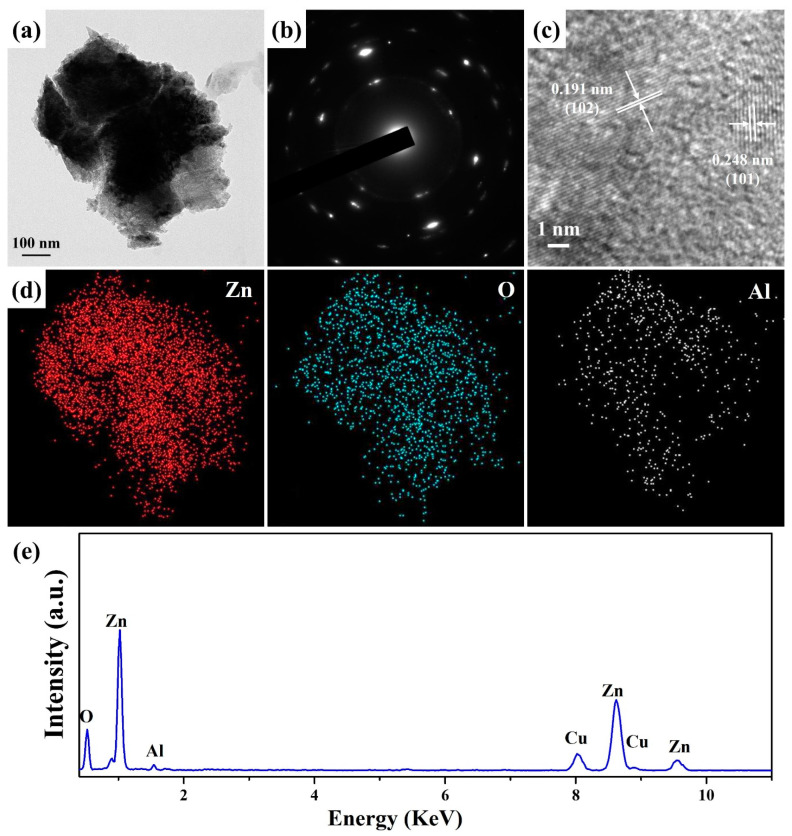
(**a**) FETEM, (**b**) SAED pattern, (**c**) HRTEM, (**d**) EDS mapping images, and (**e**) EDS spectrum of a ZnO nanowall (50 mM).

**Figure 7 materials-15-09008-f007:**
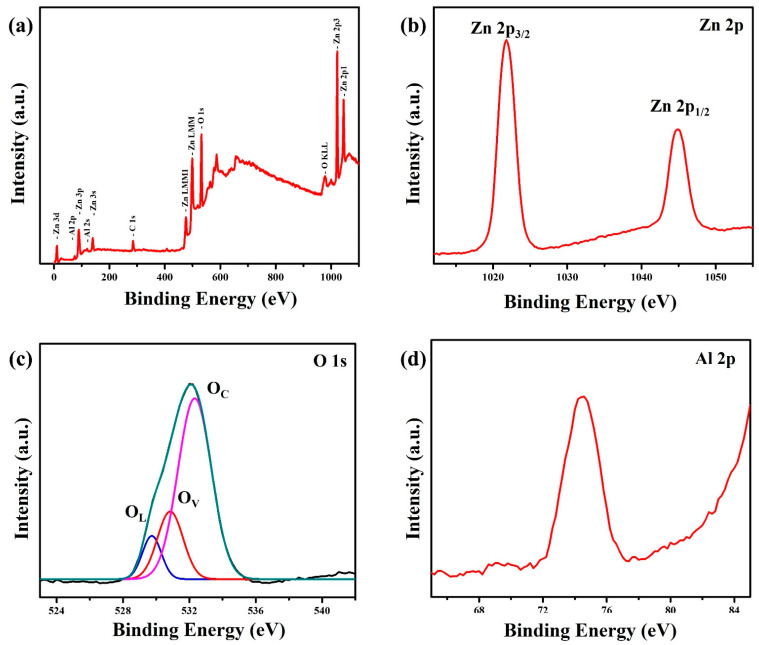
XPS spectra of the ZnO nanowalls: (**a**) survey spectrum, (**b**) Zn 2p, (**c**) O 1s, and (**d**) Al 2p.

**Figure 8 materials-15-09008-f008:**
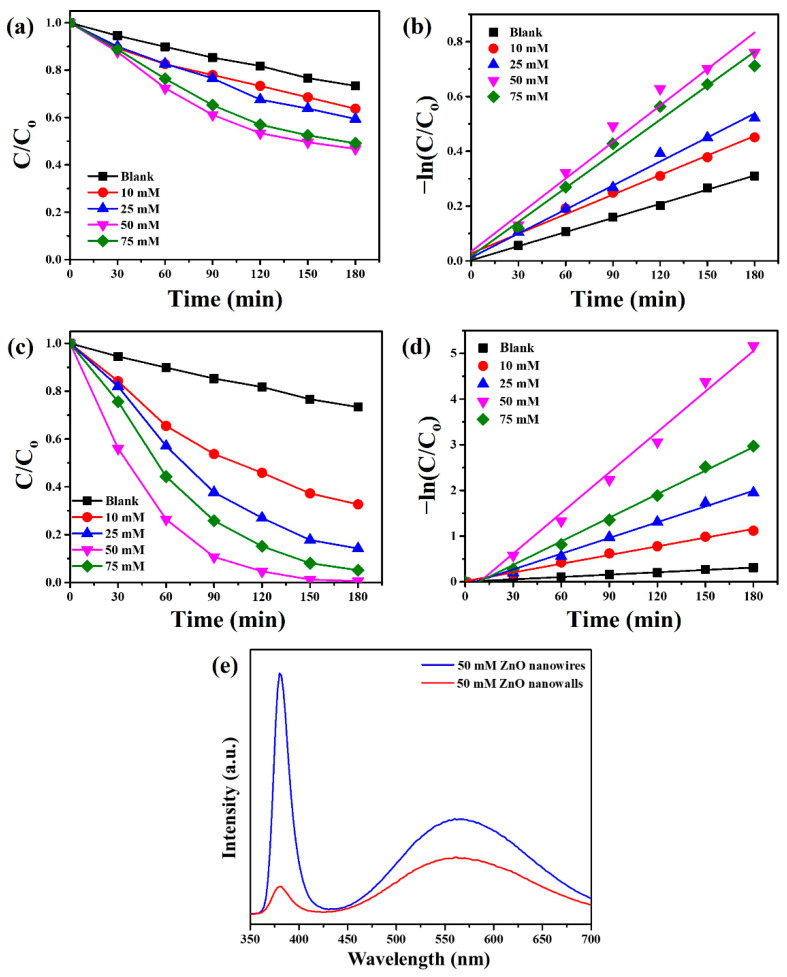
(**a**) Photocatalytic activities and (**b**) kinetic linear simulation curves of ZnO nanowires synthesized on the glass substrate with a ZnO seed film at different concentrations of ZnO precursors under UVC light irradiation. (**c**) Photocatalytic activities and (**d**) kinetic linear simulation curves of ZnO nanowalls grown on the Al foil at different concentrations of ZnO precursors under UVC light irradiation. (**e**) CL spectra of ZnO nanowires and ZnO nanowalls grew at the ZnO precursors of 50 mM.

**Figure 9 materials-15-09008-f009:**
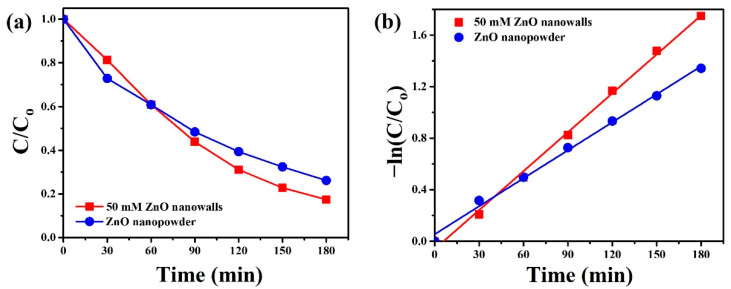
(**a**) Photocatalytic activities and (**b**) kinetic linear simulation curves of ZnO nanowalls (50 mM) and commercial ZnO nanopowder for TC solution under UVC light irradiation.

**Figure 10 materials-15-09008-f010:**
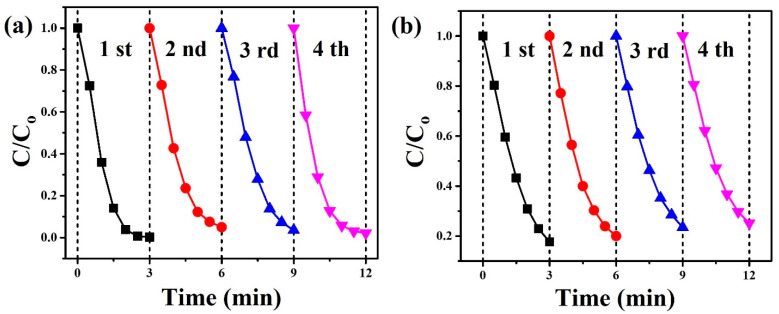
The reusability of the ZnO nanowalls (50 mM) for (**a**) MB and (**b**) TC solution under UVC light irradiation.

**Figure 11 materials-15-09008-f011:**
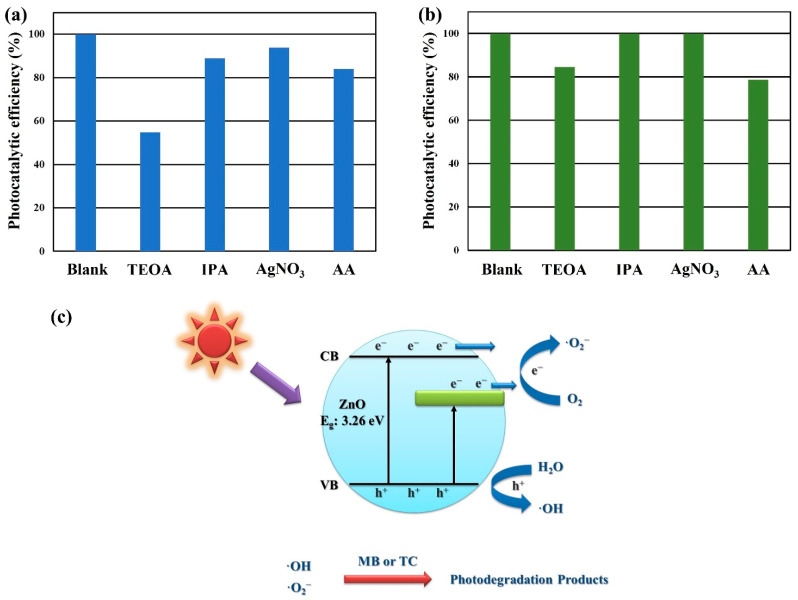
Photocatalytic activities of ZnO nanowalls for (**a**) MB and (**b**) TC solution with various scavengers under UVC light irradiation. (**c**) Schematic diagram of the electron transfer mechanism of ZnO nanowalls under UVC light irradiation.

**Table 1 materials-15-09008-t001:** The degradation of methylene blue over photocatalysts published compared to our work.

Photocatalysts	Photocatalytic Efficiency (%)	Light Sources	References
ZnO Nanowalls	99.4	UVC (5 W)	Present Work
γ-Fe_2_O_3_/Fe_3_O_4_/SiO_2_ Photocatalysts	87.5	UV (150 W)	Ref. [35]
Bi/BaSnO_3_@HNTs Nanomaterials	90.2	LED (200 W)	Ref. [36]
Cu-doped ZnO Nanoneedles	89.5	UVC (5 W)	Ref. [33]
Bi_2_WO_6_/ZIF8 Photocatalysts	85.7	Visible light (400 W)	Ref. [37]

## Data Availability

Not applicable.

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
