# Peer review of "Density-Controlled Growth of ZnO Nanowalls for High-Performance Photocatalysts"

_materials, 2022, doi:10.3390/ma15249008_

Round 1
Reviewer 1 Report
In this work a method was developed for synthesis of zinc oxide nanowalls with high photocatalytic activity in the decomposition of methylene blue and tetracycline when illuminated with ultraviolet light with a wavelength of 253,7 nm. Zinc oxide nanowires were also synthesized, while their photocatalytic activity was much lower. The morphology and structure of zinc oxide nanowires and nanowalls has been comprehensively studied. The mechanism of photocatalytic decomposition of organic substances is considered. The advantage of the developed synthesis method of zinc oxide nanowalls is simplicity, low cost and the use of available substrates.
However, there are several shortcomings in the manuscript:
1. The chemical formula of zinc nitrate dihydrate is incorrectly indicated in Fig. 1.
2. It is desirable to estimate the degree of doping of zinc oxide nanowalls with aluminum.
3. In Fig. 8, b, the black line is marked “black” instead of “blank”.
4. There is no comparison of the photocatalytic activity with the results of other authors.
Author Response
In this work a method was developed for synthesis of zinc oxide nanowalls with high photocatalytic activity in the decomposition of methylene blue and tetracycline when illuminated with ultraviolet light with a wavelength of 253,7 nm. Zinc oxide nanowires were also synthesized, while their photocatalytic activity was much lower. The morphology and structure of zinc oxide nanowires and nanowalls has been comprehensively studied. The mechanism of photocatalytic decomposition of organic substances is considered. The advantage of the developed synthesis method of zinc oxide nanowalls is simplicity, low cost and the use of available substrates.
Response: Thanks for the pertinent and positive comments.
However, there are several shortcomings in the manuscript:
- The chemical formula of zinc nitrate dihydrate is incorrectly indicated in Fig. 1.
Response: Thanks for your reminder. We have amended this mistake in new Fig. 1.
- It is desirable to estimate the degree of doping of zinc oxide nanowalls with aluminum.
Response: Thanks for your reminder. For ZnO nanowalls, the content of Al is determined to be 11.3 at% by the EDS spectrum (Figure 6e).
- In Fig. 8, b, the black line is marked “black” instead of “blank”.
Response: Thanks for your reminder. We have amended this mistake in new Fig. 8b.
- There is no comparison of the photocatalytic activity with the results of other authors.
Response: Thanks for your reminder. We have added a comparison table (Table S1) in the Supplemental Information.

Reviewer 2 Report
The authors have synthesized and studied the density-controlled growth of ZnO Nanowalls for High-Performance Photocatalysts. The work is exciting and can be considered after a few modifications to the manuscript for the readers.
Abstract: Kindly modify the abstract quantitatively by adding the detail of the ZnO morphologically and with result from photocatalyst studies.
Introduction: The section needs to bring more recent development in the synthesis of ZnO nanostructures. Here are some suggestions:
1) Vemuri, Suresh Kumar, et al. "Fabrication of silver nanodome embedded zinc oxide nanorods for enhanced Raman spectroscopy." Colloids and Surfaces A: Physicochemical and Engineering Aspects 639 (2022): 128336.
2)Le, Anh Thi, et al. "Immobilization of zinc oxide-based photocatalysts for organic pollutant degradation: A review." Journal of Environmental Chemical Engineering (2022): 108505.
Result: The section is well written.
The dimensions of ZnO nanorods need to be added in figure 2 FESEM analysis. A graph can be added to show the change in the length and diameter of nanorods with a change in the concentration.
The author can provide the cross-sectional image of the nanowall in figure 4 with all the dimensions. Kindly mention the measurements obtained from the FESEM analysis.
The authors need to add Raman spectroscopy analysis to investigate any defect presence during synthesis.
A comparative study or table with reported literature needs to include to demonstrate the novelty and the better efficiency of the nanostructure.
Conclusion: Kindly modify the section more quantitatively.
A few grammatical errors can be corrected in the revised manuscript.
Author Response
The authors have synthesized and studied the density-controlled growth of ZnO Nanowalls for High-Performance Photocatalysts. The work is exciting and can be considered after a few modifications to the manuscript for the readers.
Response: Thanks for the pertinent and positive comments.
Abstract: Kindly modify the abstract quantitatively by adding the detail of the ZnO morphologically and with result from photocatalyst studies.
Response: Thanks for your reminder. We have amended this description in the revised manuscript.
Introduction: The section needs to bring more recent development in the synthesis of ZnO nanostructures. Here are some suggestions:
1) Vemuri, Suresh Kumar, et al. "Fabrication of silver nanodome embedded zinc oxide nanorods for enhanced Raman spectroscopy." Colloids and Surfaces A: Physicochemical and Engineering Aspects 639 (2022): 128336.
Response: Thanks for your reminder. We have updated this reference in the revised manuscript.
2)Le, Anh Thi, et al. "Immobilization of zinc oxide-based photocatalysts for organic pollutant degradation: A review." Journal of Environmental Chemical Engineering (2022): 108505.
Response: Thanks for your reminder. We have updated this reference in the revised manuscript.
Result: The section is well written.
Thanks for the pertinent and positive comments.
The dimensions of ZnO nanorods need to be added in figure 2 FESEM analysis. A graph can be added to show the change in the length and diameter of nanorods with a change in the concentration.
Response: Thanks for your reminder. We have added a comparison figure of the diameters and lengths of ZnO nanowires under the different concentrations of ZnO precursors (Figure S1) in the Supplemental Information.
The author can provide the cross-sectional image of the nanowall in figure 4 with all the dimensions. Kindly mention the measurements obtained from the FESEM analysis.
Response: Thanks for your reminder. We have added the cross-sectional FESEM images of ZnO nanowalls synthesized at the ZnO precursors concentration of 50 and 75 mM (Figure S1) in the Supplemental Information. However, the zinc oxide nanowalls at 10mM and 25mM are too loose. They are easily broken during the fragmentation process induced that cannot be clearly observed.
The authors need to add Raman spectroscopy analysis to investigate any defect presence during synthesis.
Response: Thanks for your reminder. Due to the failure of the light source of the Raman spectroscopy, it takes a period to repair before the measurement can be carried out.
A comparative study or table with reported literature needs to include to demonstrate the novelty and the better efficiency of the nanostructure.
Response: Thanks for your reminder. We have added a comparison table (Table S1) in the Supplemental Information.
Conclusion: Kindly modify the section more quantitatively.
Response: Thanks for your reminder. We have amended this description in the revised manuscript.

Round 2
Reviewer 2 Report
The authors have made significant modification to the manuscript and thus it can be acceptable in present form.